# Combined Treatment (Alkali + Thermal) of *Calotropis procera* Fiber for Removal of Petroleum Hydrocarbons in Cases of Oil Spill

**DOI:** 10.3390/polym13193285

**Published:** 2021-09-26

**Authors:** Raoni Batista dos Anjos, Larissa Sobral Hilário, Aécia Seleide Dantas dos Anjos, Emily Cintia Tossi de Araújo Costa, Tarcila Maria Pinheiro Frota, Amanda Duarte Gondim, Djalma Ribeiro da Silva

**Affiliations:** Nucleus of Primary Processing and Reuse of Produced Water and Waste, Federal University of Rio Grande do Norte, Natal 59078-970, RN, Brazil; larissasobralhilario@yahoo.com.br (L.S.H.); aeciadantas@gmail.com (A.S.D.d.A.); emily.tossi@gmail.com (E.C.T.d.A.C.); tarcilafrota10@gmail.com (T.M.P.F.); amandagondim.ufrn@gmail.com (A.D.G.); djalmarib@gmail.com (D.R.d.S.)

**Keywords:** sorption, petroleum, paraffin, oil spill, experimental planning, *Calotropis procera*

## Abstract

The objective of this study was to evaluate the *Calotropis procera* fiber treated with NaOH combined with heat treatment as sorbent material for removal of petroleum and derivatives in cases of oil spill. The effects of oil viscosity, fiber/oil contact time, and the type of sorption system (oil and oil/water) were evaluated by experimental planning. The fiber obtained was characterized by Fourier transform infrared spectroscopy (FTIR), field-emission scanning electron microscopy (SEM-FEG), thermogravimetric analysis (TG/DTG), contact angle, and wettability. The fiber treated by combining NaOH and heat treatment (CPNaOHT) exhibited a large structure with an internal diameter of 42.99 ± 3.98 μm, roughness, and hydrophobicity on the surface with a contact angle of 101 ± 2°. The sorption capacity of oil ranged from 190.32 g/g to 98.9 g/g. After five cycles of recycling, the fiber still maintains about 70% of its initial sorption capacity and presented low liquid desorption (0.25 g). In this way, it can be used as an efficient sorbent to clean up spills of oil and oil products.

## 1. Introduction

Although statistics show a decrease in the number of accidents involving oil spills at sea, they continue to occur, threatening the environmental quality of coastal ecosystems such as beaches, rocky shores, and mangroves, among others [1,2]. In 2019, on the coast of northeastern Brazil, an oil spill reached a coastal strip of 4334 km in 11 states in the Northeast and Southeast, reaching 120 municipalities and 724 localities [3,4].

In accidents with this level of commitment, it is necessary to establish and employ efficient cleaning procedures. Therefore, the choice of procedures is essential to minimize impacts and accelerate the recovery processes of contaminated environments [5].

The most common oil spill containment method is the use of barriers containing sorbent materials with characteristics of high mechanical resistance and adequate physicochemical properties, such as polyurethane foams and polypropylene fibers. These materials, which have a high production cost and are non-biodegradable, have been widely used in the remediation of oil spill scenarios due to their high oil retention capacity, which can be more than 70 times the weight of the material itself [6,7,8,9]. Given this scenario, there is a demand for efficient and low-cost renewable materials for oil removal in cases of oil spills.

Rice husks, mandacaru, sawdust, and sugarcane bagasse, among other biomasses, have the advantages of low cost, easy availability, and biodegradability [10,11,12]. However, natural organic materials have the disadvantages of low selectivity and oil sorption efficiency. Some vegetable fibers with a tubular structure such as cotton, painera, cattail, and kapok fiber have shown greater oil sorption capacity than other natural organic materials and, recently, vegetable fiber from *Calotropis procera* [13,14,15,16].

The *Calotropis procera* (Aiton) W. T (CP) is a species originally from Africa and Asia, and therefore considered an invasive species in Brazil [17], and which can currently be found in several regions of the country. However, it is in the Northeast where large populations of the species are established [17,18,19,20,21]. It is a polyvalent plant, which can be used for medicinal, forage, and fuel purposes, wood and fiber production, phytoremediation, synthesis of nanoparticles, and biosorbents [19]. The species has gained prominence due to its good water repellency, high oil selectivity, hollow structure, and a wax layer on the surface, making it a potential raw material for manufacturing high-performance sorbents for removing oil from water [14,15,16,17]. Several researchers have published papers attesting to the sorption potential of *Calotropis procera* fiber oils, such as Thilagavathi et al. (2018) indicating a sorption capacity of 23–40 g/g, Hilário et al. (2019) of 76–181 g/g, and Anjos et al. (2020) of 92–104 g/g [14,15,16].

To improve the intrinsic properties or change the characteristics of natural fibers and biomass, these can be subjected to chemical and physical treatments, obtaining materials with new hydrophobic-oleophilic characteristics and high oil sorption capacity [15]. In previous works, Hilário et al. (2019) treated the CP fiber thermally, which showed an increase in oil sorption capacity ranging from approximately 35% to 137% [14], and Anjos et al. (2020) using the alkaline treatment with NaOH in the CP fiber, showed an increase in the internal diameter and gain in the surface area of the fiber, improving the oil sorption capacity by 68% when compared to the fresh fiber [15]. However, there are no studies of *Calotropis procera* fiber treated combining alkaline and thermal treatments for oil removal. Therefore, the present study aims to evaluate the combined NaOH and thermal treatment in *Calotropis procera* fiber to improve the oil sorption capacity by modifying the surface and the hollow structure of the fiber. Several parameters such as oil viscosity, fiber/oil contact time, and type of sorption system were the target of this study.

## 2. Materials and Methods

### 2.1. Material

To carry out this study, the fruits of *Calotropis procera* (Ait.) W. T were collected in the city of Natal, Rio Grande do Norte, Brazil. Paraffin (Density at 25 °C, 0.766 g/cm^3^ and viscosity at 25 °C, 2.25 cP), crude oil (Density at 25 °C, 0.861 g/cm^3^ and viscosity at 25 °C, 73.6 cP), and marine diesel samples (Density at 25 °C, 0.825 g/cm^3^ and viscosity at 25 °C, 2.38 cP) used in the sorption tests were supplied by an petroleum company located in Rio Grande do Norte, Brazil. Diesel sample (Density at 25 °C, 0.813 g/cm^3^ and viscosity at 25 °C, 1.95 cP) was purchased at a fuel station in the city of Natal, Rio Grande do Norte, Brazil. The reagent water used in the tests was obtained through a water ultrapurifier, model Integral 5, Millipore, city of Bedford, MA, USA. Sodium hydroxide PA ACS (NaOH) was purchased from Vetec, Sigma Aldrich, city of Duque de Caxias, Rio de Janeiro, Brazil and was used for alkaline treatment.

### 2.2. Combined Treatment of Calotropis Procera

Previously the fibers of *Calotropis procera* were removed from the interior of the fruits and manually separated from the seeds and dried at room temperature (25 ± 1 °C) for 24 h. Then, 2 g of fiber were placed in a beaker in a 10% (*v/v*) NaOH solution, under stirring for 1 h, according to Anjos et al. (2020) [15]. After treatment with NaOH the fiber was washed with ultrapure water and taken to a muffle for thermal treatment, as reported by Hilário et al. (2019) [14], modifying the duration of treatment from 1 h, 2 h, and 3 h [22]. After the treatments, the fibers obtained were cooled in a desiccator and stored in zip lock bags.

### 2.3. Determination of the Sorption Capacity

The assessment of oil sorption capacity was performed using the methodology reported by Anjos et al. (2020) [15]. An amount of 0.01 g of the fiber was immersed in the systems (O and O/W). Then, the swollen fibers were removed and placed to drain the excess in a stainless steel screen for 5 min. The oil sorption capacity (g/g), *S*, was determined by the gravimetric method and represents the proportion of oil mass to dry fiber mass, calculated by Equation (1):(1)$$S=\frac{w_{s1}-w_{s0}-w_{s2}}{w_{s0}}$$
where *W_S_*_0_ is the dry fiber mass (g), *W_S_*_1_ is the swollen fiber mass after oil absorption (g), and *W_S_*_2_ is the mass of water sorbed by the fiber (g). The sorption tests were performed at room temperature (25 ± 1 °C) and all weighings used an analytical balance (0.001 g).

### 2.4. Design of Experiments

In order to evaluate the effects of oil viscosity, fiber/oil contact time and the type of sorption system with only oil (O), simulating an oil spill on land, and oil/water (O/W), simulating an oil spill in water, on the sorption capacity of the CPNaOHT treated fiber, a statistical modeling was performed using the 2^3^ factorial design matrix. Establishing these three factors as independent variables: oil viscosity, fiber/oil contact time, and type of sorption system, and as a dependent variable, oil sorption capacity (g oil/g fiber). For each factor, two levels were selected: low (−1) and high (+1), as shown in Table 1. The tests were performed in duplicates, in a total of 16 experiments.

To assess the statistical significance of the models, the analysis of variance (ANOVA) study was performed with 95% confidence. The calculated F value (F_calc_) was compared to the tabulated F value (F_tab_) for the F distribution with the respective degrees of freedom [23]. The polynomial equation of the second-order (Equation (2)) expressed below was used to correlate the oil sorption capacity and the dependent variables. All analysis, calculations, and statistical graphics were obtained with the STATISTICA 7.0 software.
(2)$$y=b_{0}+\overset{k}{\underset{i=1}{\sum }}b_{i}x_{i}+\overset{k}{\underset{i=1}{\sum }}b_{i}x_{i}^{2}+\overset{k-1}{\underset{i=1}{\sum }}\overset{k}{\underset{j=i+1}{\sum }}b_{ij}x_{i}x_{j}$$
where, *I* = 1–3 and *j* = 1–3.

### 2.5. Recycle

To evaluate the recycling of fiber treated CPNaOHT, sorption with petroleum was performed [14,15]. A total of 10 mg of the fiber was immersed in 5 mL of petroleum for a time of 60 min at room temperature (25 ± 1 °C). After this time of fiber/oil contact, they were compressed with the aid of tweezers, and their recycling capacity was calculated from the ratio of reabsorption mass by the initial sorption mass.

### 2.6. Desorption

After the immersion time of the fiber CPNaOHT in oil with a contact time of 1 h, drainage was performed on a stainless steel screen for pre-determined times of 1 min; 5 min, 10 min, 20 min, 30 min, 40 min, and 60 min [24,25]. The results are presented in terms of oil mass impregnated in the sorbent as a function of time. The experiments were performed in triplicates.

### 2.7. Characterizations

The surface morphology of the treated fibers was characterized using a scanning electron microscope with field emission (Auriga 40, Zeiss, Oberko-chen, Germany). To obtain the FTIR spectra of the samples, they were recorded on a Frontier spectrometer (Perkin Elmer, Waltham, MA, USA), with a horizontal attenuated total reflectance (ATR) accessory. The surface wettability of fibers was evaluated by measurement of water and diesel contact angles, using a Tensiometer, model K100C (Krüss, Hamburg, Germany). The thermal stability of the CP, CPNaOH, and CPNaOHT was evaluated by thermogravimetry analyses-(TG) and Derivative Thermogravimetry using a thermogravimetric analyzer from NETZSCH, TG209F1 Libra (Netsch, Selb, Germany). Approximately 7 mg of sample were used in the TG/DTG analyses, heating rate 10 °C.min^−1^, temperature range from 28 to 600 °C, under a dynamic oxygen atmosphere and flowrate of 20 mL.min^−1^.

## 3. Results and Discussion

### 3.1. Combined Treatment (Alkali + Thermal)

The *Calotropis procera* fiber was initially treated with a NaOH solution (10% *v/v*) for 1 h [23], then it was taken to a muffle for thermal treatment at temperatures of 150 °C and 200 °C with heating rate of 10 °C.min^−1^, varying the duration of the heat treatment in 1 h, 2 h, and 3 h [14,22]. To evaluate the effects of the treatments, sorption tests were carried out with only paraffin for the oil only (O) system, with a fiber/oil contact time of 1 h. The tests were carried out in triplicates. Figure 1 shows the sorption capacities for CP fiber treated at temperatures of 150 °C and 200 °C with the duration of treatments from 1 h to 3 h, for the paraffin tests.

It is observed in Figure 1 that increasing the treatment temperature from 150 °C to 200 °C increased the paraffin sorption capacity from 30% to 57% for all the duration of the heat treatment. Hilário et al. (2019) also observed an increase in the CP sorption capacity with an increase in the treatment temperature from 150 °C to 200 °C, with the treatment at 200 °C showing the best CP sorption capacity for crude oil [14]. For comparison purposes, the paraffin sorption test for CP in nature was performed, which showed sorption of 53.4 g/g. Where the fibers treated with NaOH and thermally at 150 °C and 200 °C showed an increase in paraffin sorption capacity from 52% to 74% compared to the untreated fiber.

The effect of the duration of the heat treatment on the oil sorption capacity of the CP fiber was also evaluated (Figure 1). Observing the increase in the duration of heat treatment from 1 h to 3 h, the sorption capacity of the treated fibers decreased, with losses ranging from 5 to 34%. It was also observed that the fibers after the treatment duration time of 2 h and 3 h proved to be brittle, possibly due to fiber degradation with an increase in the treatment duration time. This may have led to a decrease in the paraffin sorption capacity. Husseien et al. (2009) observed that increasing the duration of the heat treatment from 1 h to 3 h at 200 °C increased the sorption capacity of the corn stem for the diesel and dye studied [22]. In Figure 2, it is possible to observe the visual aspects of the fibers treated at 150 °C and 200 °C, for the duration of heat treatment of 1 h, 2 h, and 3 h.

The fiber treated with NaOH (10% *v/v*) and thermally at 200 °C for 1 h had the highest sorption capacity of 95.94 ± 0.88 g/g (Figure 1), being chosen for the other tests of sorption and characterization. The treated fiber chosen was named CPNaOHT.

### 3.2. Design of Experiments

The levels of significant factors and effects of interactions between the factors that were analyzed by factorial design 2^3^. The three factors: viscosity (cP) of the oil, fiber/oil contact time (h), and the type of oil-only sorption system (O), and oil and water (O/W) were selected according to the literature [14,15,24,25,26,27,28] and planned according to the matrix described in Table 2. For modeling purposes, the independent variables (factors) were scaled for coded variables (Table 2), and the sorption capacity of the materials for oils was named *S*_1_ (g/g).

To assess the statistical significance of the model, the analysis of variance (ANOVA) study was performed at 95% confidence. According to the analysis of variance performed for the results obtained, it was verified that the model is significant with a correlation coefficient of R^2^ de 0.99992, R-adj de 0.99985 e *p* < 0.05 (Table 3). The value of the F_cal_ test was compared to the value of the F_tab_ test for the distribution F with the respective degrees of freedom and the value of F_cal_ regression was 4113 times greater than the F_tab_, at a confidence level of 95%. These facts show that the above model explains 99.9% of the variation of the experimental data indicating that the model is meaningful and predictive, so the model can be applied to predict the oil sorption capacity within the ranges discussed in this work [23].

Based on the experimental design shown in Table 3, the regression model was constructed to describe the functional relationship between the variables (independent variables) and CPNaOHT sorption capacity (response). The polynomial regression model of second-order in terms of coded variables (*X*_1_, *X*_2_, and *X*_3_) that predict the sorption-city cover (*S*_1_) of CPNaOHT can be written by Equation (3), considering the significant factors (*p* < 0.05) and their interactions. The equation was obtained using Statistica software version 7.0.
(3)$$S_{1}=111.8306+20.0994X_{1}+9.1331X_{2}+12.4994X_{3}+9.1331X_{2}+14.5494X_{1}X_{2}+12.6481X_{1}X_{3}+3.8494X_{2}X_{3}+5.7106X_{1}X_{2}X_{3}$$

The model fit can be observed in Figure 3a through the relation of experimental and predicted values for CPNaOHT oil sorption. For model validation purposes, a new oil with a viscosity of 2.38 cP (Marine diesel) was chosen to perform a sorption test at the fiber/oil contact time of 1 h, using the sorption system with only oil (O). The experiments were carried out in triplicate and the mean sorption capacity determined experimentally was 94.65 ± 0.22 g/g. The value calculated by the model was 95.40 g/g, with a low absolute error of 0.75. All these statistical estimators reveal that the oil sorption model (*S*_1_) is statistically validated to predict the response in the experiment region.

### 3.3. Effect of Variables on CPNaOHT Sorption Capacity

Whereas the model has been adjusted (meaningful and predictive), the effects of variables and interactions between them over the CPNaOHT sorption capacity were evaluated based on the Pareto diagram (Figure 3b), to a confidence level of 95%. According to the Pareto diagram (Figure 3b), all effects were significant (*p* < 0.05), alone or combined for CPNaOHT sorption capacity. Evaluating the effect of isolated variables, it is observed that the viscosity showed a greater influence on the sorption process. The oil viscosity is a parameter of great importance in the sorption process as reduced oil viscosity reduces sorption into the pores and vessels capillaries of materials, and more viscous oils have higher sorption due to adhesion to surfaces of materials and in pores [24,25,26,27,28]. Wei et al. (2003) showed in their experiments that increasing the viscosity from 18.7 cP to 54.8 cP contributes significantly to the increase in the sorption capacity of polypropylene [24]. However, it is noted that even for low viscosity oils (cP = 2.25), fiber CPNaOHT has a high average sorption capacity of 98.86 g/g, demonstrating the potential of CPNaOHT for removal/sorption of oils with viscosity varying from 2.25 to 73.6 cP.

The type of sorption system (O and O/W) had a significant influence on the sorption capacity CPNaOHT (*p* < 0.05), the sorption system O/W which had the highest oil removal capacity. This is possibly due to the difference in polarity between oil and water, causing the oil to be pushed into the fiber lumens by the repulsion forces added to the capillary force.

The contact time between fiber and oil in the sorption process showed a directly proportional effect (Figure 3b), this implies that the higher the CPNaOHT contact time with the oil, the greater the sorption capacity. The increase in contact time led to decreased sorption of paraffin, possibly by lower viscosity paraffin, having full capacity in just one hour. However, the oil has a lower sorption rate within the pores and capillaries in the 1 h contact time, reaching full capacity within 24 h.

In Figure 4, it is possible to observe the effects of the independent variables (viscosity, system and contact time) in relation to the average values of the oil sorption capacity within the 95% confidence interval, for the set of experiments performed according to with the experimental design (Table 2). It is observed that all variables showed an increase in the sorption capacity with the increment from −1 to 1, with the sorption condition being for the oil with higher viscosity (73.6 cP), the O/W system and the longest oil contact time/fiber (24 h), which had the best oil removal capacity. The results show that CPNaOHT fiber has greater potential for sorption and use in case of oil spills in aquatic environments than in terrestrial environments (without the presence of water), reaching a removal of 84.31 g/g to 190.32 g/g of oil in water.

### 3.4. Recycle Desorption

To assess the ability of recycling/reuse CPNaOHT has performed a sorption system with only oil (O) to the time of contact fiber/oil 60 min, followed by simple compression and weighed again for 5 cycles. The reuse of the fibrous oil sorbent in terms of sorption capacity during five cycles of sorption/desorption is shown in Figure 5a. It is observed the decrease of sorption capacity for each 90% duty cycle, 82%, 71%, 62%, and 49%, but the average capacity of resorption of the fibers after 5 cycles was higher than 70% oil as compared to the sorption initial, demonstrating that CPNaOHT fiber still has the potential for reuse even after sorption 5 consecutive uses.

The study of liquid loss from sorbents over time is of paramount importance in actual application [24,25]. Figure 5b shows the desorption of petroleum by CPNaOHT in times from 1 to 60 min, where a general trend of net loss divided into two stages is observed. Initially, the liquid loss rate is higher and the holding capacity drops. A transition zone, where desorption virtually ends, is observed, reaching a steady-state condition under which no liquid is lost due to dripping (Figure 5b). The maximum loss of oil mass by desorption of CPNaOHT was approximately 0.25 g, after 60 min of drainage.

### 3.5. Short Time Sorption Tests

In real cases of oil spills, the faster the process of sorption occurs, the smaller the environmental impacts on the marine environment or land. Anjos et al. (2021) evaluated the concentrations of BTEX (benzene, toluene, ethylbenzene, and xylenes) in the soluble fraction of oil, simulating a spill by varying the contact time of the oil in seawater from 1 to 24 h, where the concentrations of benzene, carcinogenic compound, and mutagenic exceeds the most restrictive value by CONAMA (51 µg/L) for saline waters where there is fishing or cultivation of organisms, for intensive consumption purposes (BRASIL, 2005), in just 1 h [28,29]. With 24 h of oil spill into seawater, the average concentrations of benzene, toluene, ethylbenzene, and xylenes were 1409 µg/L, 1279 µg/L, 67 µg/L, and 357 µg/L, respectively, showing the importance of the removal of this oil in a short period. To evaluate the sorption capacity of CPNaOHT in shorter times, sorption tests were carried out for short times of 5, 20, 40 min for the two sorption systems with only oil and oil/water, for petroleum and paraffin [28].

Figure 6 shows the average sorption capacity for short time tests (5, 20, and 40 min) for systems with only oil and oil/water. Paraffin showed a characteristic of increasing sorption capacity with increasing time from 5 to 40 min, for both sorption systems. Petroleum presented a stable sorption capacity already at 5 min, remaining up to 40 min, for the oil-only system, and in the oil/water system, it presented an increase in sorption capacity over time. The short sorption time tests showed that the CPNaOHT fiber showed good oil removal capacity, ranging from 74.1 g/g to 102.9 g/g, which can be a good alternative in emergency cases of oil containment.

The data from the present study indicate that the fiber treated by combining NaOH and heat treatment (CPNaOHT) showed an increase in petroleum sorption capacity of 150% and 74% for paraffin when compared to CP fiber in nature. The treated fiber had a higher sorption capacity than CPT150 (thermal treatment at 150 °C), CPT200 (thermal treatment at 200 °C), CPNaOH (treated with NaOH), CPNaClO_2_ (treated with NaClO_2_), and CPHT (treated with hydrothermal) fibers, previously studied by Hilário et al. (2019) and Anjos et al. (2020), suggesting that the new treatment process significantly improved the sorption capacity of the fibers and could be used as an efficient sorbent for the removal of spilled oil in both aquatic and terrestrial environments [14,15]. Figure 7 represents the sorption process of CPNaOHT fiber in the O/W system, using diesel oil (Density at 25 °C, 0.813 g/cm^3^ and viscosity at 25 °C, 1.95 cP), also showing that the fiber has good sorption capacity for diesel (Figure 7b–d) and good buoyancy in water (Figure 7a) (A video file of the CPNaOHT fiber sorption test with diesel oil was provided in Appendix A).

### 3.6. Characterizations

The morphology of CPNaOHT obtained was characterized by SEM-FEG and the micrographs are shown in Figure 8a,c. The CP and CPNaOH fibers previously studied by Hilário et al. (2019) and Anjos et al. (2020), respectively, presented a cylindrical shape with large lumens, thin and smooth walls [14,15]. After CP was combined with NaOH and thermal treatment at 200 °C for 1 h, the fiber surface became rough (Figure 8b). Previous treatment with NaOH may have left the fibril structure more exposed and susceptible to the action of heat treatment, thus generating surface roughness [30]. It was also possible to observe that the internal diameter increase after CPNaOHT treatment was 80% (42.99 ± 3.98 µm, *n* = 10), compared to CP fiber without treatment. The CP fiber treated only with NaOH showed an internal diameter increase of 57% (37.47 ± 3.80 µm), when compared to the CP fiber (23.84 ± 4.44 µm) [26]. Such a hollow tubular structure combined with roughness increases the CPNaOHT sorption potential of various oils.

To verify the hydrophobic and oleophilic properties of the CPNaOHT fiber, the wettability test in water and oil on the fiber surface was performed, as shown in Figure 8b, where the hydrophobicity and olefinicity are visible on the CPNaOHT surface. The contact angle (θ) for CPNaOHT water was 101 ± 2° (Figure 9a,b), showing that the material is hydrophobic, that is, oil is preferred over water. Comparing the contact angle of CPNaOHT with fiber treated only with NaOH (CPNaOH θ = 114°) (Anjos et al., 2020) and CP fiber in natura (θ = 128°) (Hilário et al., 2019), it was is possible to observe that the surface of the material after the combined treatment was modified, sorbing more water [14,15]. The use of plant materials as sorbents in an aqueous medium generally present high water sorption (Annunciation et al., 2005), however in the present study the amount of water sorbed in relation to oils was negligible (Figure 9c), ranging from 0.28 to 0.70 g/g in the range of 5 to 60 min [11]. It is also observed that water sorption increased with time. The CPNaOHT fiber sorbed more water when compared to the CP fiber in nature and the fiber treated only with NaOH (CPNaOH), possibly by removing the wax from the CP surface by the combined treatment. The results of the contact angle and sorption for water suggest the hydrophobic character of the CPNaOHT fiber.

Figure 10 shows the FTIR spectra of the CPNaOHT, CP, and CPNaOH fiber [14,15]. When comparing the spectra of CP in nature and treated only with NaOH (CPNaOH) there was no significant change in the spectra, however in the region of the functional groups’ CH (2920 cm^−1^), C=O (1734, 1368, and 1244 cm^−1^), and CO (1032 cm^−1^) [14], there was a decrease in intensities possibly associated with removal of wax on the fiber surface with NaOH treatment [15]. In CPNaOHT, it was noted attenuation of peaks close to lignin (1505 and 1597 cm^−1^) and hemicellulose (1734 and 1244 cm^−1^) [15]. The peak of 897 cm^−1^ was also attenuated, possibly related to the removal of hemicellulose with the combined alkaline plus thermal treatment. Also, there was a minimization of bands in the region between 1602 and 1508 cm^−1^.

When comparing the CP fiber spectra with CPNaOHT, the heat-treated sample showed the decreased intensity of functional groups, including CH (2915 cm^−1^), C=O (1734, 1368 and 1244 cm^−1^), and CO (1032 cm^−1^) [14]. Also being noted, decrease of peaks close to lignin (1505 and 1597 cm^−1^) and hemicellulose (1737 and 1248 cm^−1^) [15,31].

To evaluate the thermal stability of CPNaOHT, thermogravimetric studies (TG/DTG) were performed. Figure 11 shows the TG/DTG curves for CP, CPNaOH, and CPNaOHT [25].

In Figure 11 we can see an initial mass loss for CP, CPNaOH, and CPNaOHT that occurred below 100 °C due to the release of weakly bound water molecules. It is observed that CPNaOHT presented greater loss of weakly bound water when compared to CP and CPNaOH (Figure 11). The second event (Figure 10a) shows a sudden mass loss, referring to hemicellulose degradation [31], similar for CP, CPNaOH, and CPNaOHT. The DTG results indicate a significant decrease in the decomposition temperature of CPNaOHT fiber hemicellulose. This is possibly due to exposure of fibrils to heat treatment after pretreatment with NaOH. The same happened in the region between 250 and 400 °C with the cellulose and/or lignin degradation peaks. For CPNaOH between 350 and 500 °C, a single peak was observed, which can be attributed to lignin degradation by the alkali treatment. The fiber treated by combining NaOH and thermal treatment showed a reduction in degradation temperature.

## 4. Conclusions

The combined treatment (alkaline and thermal) in *Calotropis procera* fiber significantly improved the oil sorption capacity. It was found that the treatment significantly increased the hollow space of the fiber, there was a roughness gain on the fiber surface and a contact angle with water of 101 ± 2° and 0° for oil, showing potential for hydrocarbon sorption, hydrophobicity, and great affinity for oil. Oil sorption capacities ranged from 84.31 to 190.32 g/g of paraffin and/or petroleum. The recycle test demonstrates a drop in the original oil sorption capacity, reaching recovery up to 70% after five reuse cycles. Through experimental planning, it was possible to evaluate the effect of viscosity, fiber/oil contact time, and type of sorption system on the sorption capacity of CPNaOHT, revealing that fiber has greater potential for sorption and use in case of oil spills in aquatic environments than in terrestrial environments (without the presence of water). Short-term trials demonstrated that CPNaOHT fiber has the potential for use in emergency cases. In conclusion, the treated fiber has excellent environmental compatibility and reuse, and it can be used as a sorbent for oil removal in spills of oil and its derivatives. Further investigations of CP and CPNaOHT fibers are planned and large-scale fabrication will be explored.

## Figures and Tables

**Figure 1 polymers-13-03285-f001:**
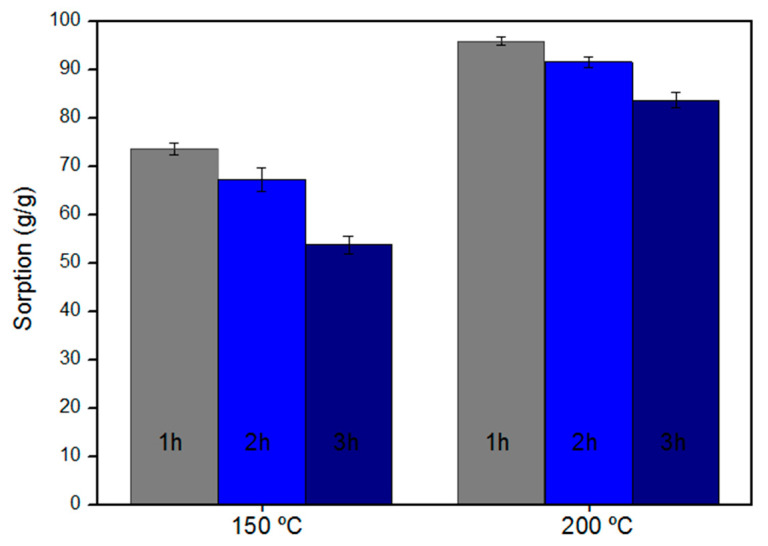
Paraffin sorption capacity of CP fiber treated at 200 °C and 150 °C varying the duration of the heat treatment in 1 h, 2 h, and 3 h, for the fiber/oil contact time of 1 h.

**Figure 2 polymers-13-03285-f002:**
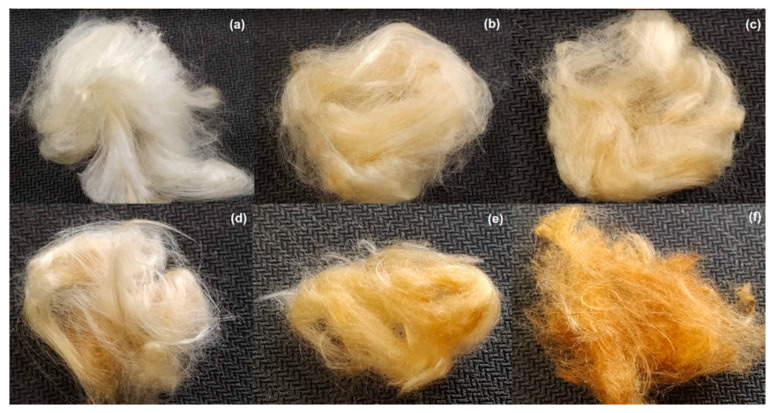
Image of fibers treated whit NaOH and at 150 °C for (**a**) 1 h, (**b**) 2 h, (**c**) 3 h and 200 °C for (**d**) 1 h, (**e**) 2 h, (**f**) 3 h.

**Figure 3 polymers-13-03285-f003:**
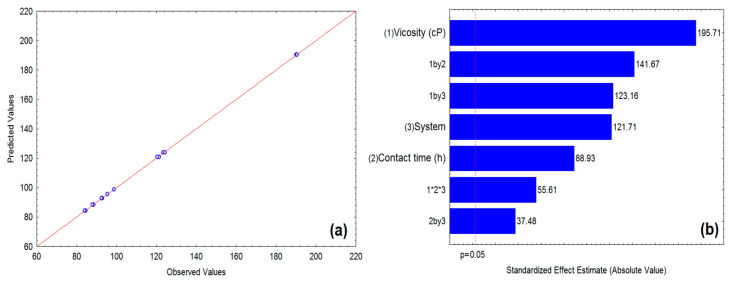
Relationship between experimental and predicted values for oil sorption (**a**) and Pareto diagram (**b**).

**Figure 4 polymers-13-03285-f004:**
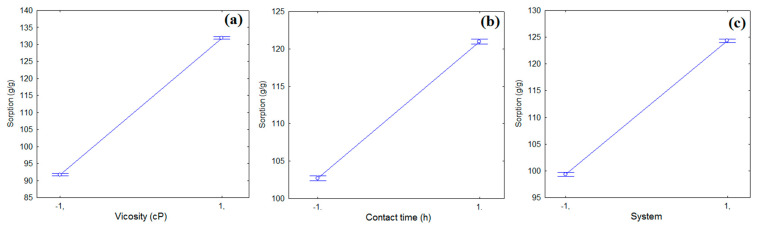
Effects of the independent variables: (**a**) viscosity, (**b**) contact time, and (**c**) sorption system, in relation to the mean values of the oil sorption capacity (95% confidence interval).

**Figure 5 polymers-13-03285-f005:**
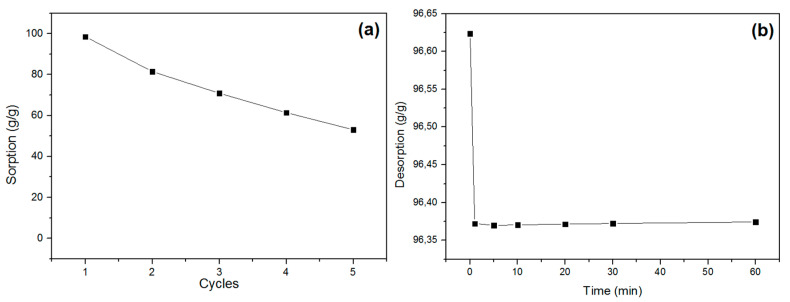
Recycling (**a**) and desorption (**b**) tests for CPNaOHT using petroleum.

**Figure 6 polymers-13-03285-f006:**
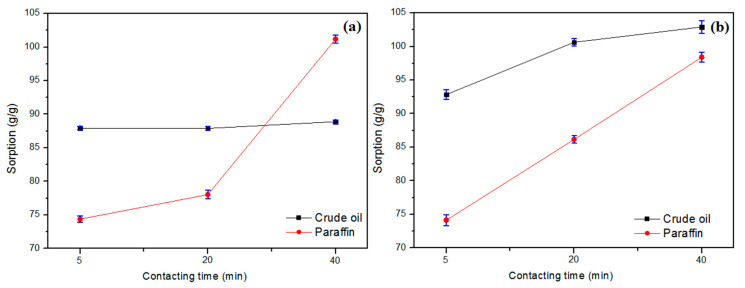
Short-time sorption tests for system (**a**) oil only and (**b**) oil/water, for petroleum and paraffin.

**Figure 7 polymers-13-03285-f007:**
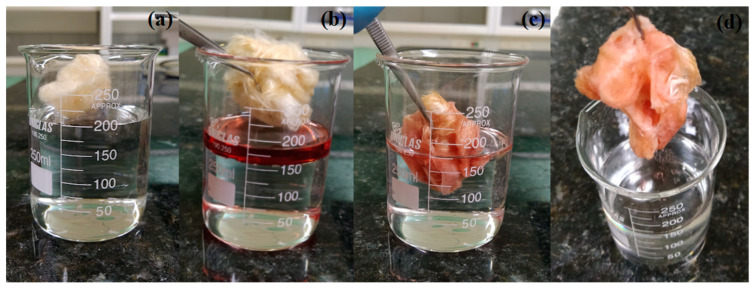
Digital images of the water sorption test (**a**) and removal of diesel oil in water using CPNaOHT (**b**–**d**), (**b**) a volume of oil was added in distilled water forming a supernatant layer, (**c**) rapid oil sorption diesel by CPNaOHT, and (**d**) clean water after diesel sorption.

**Figure 8 polymers-13-03285-f008:**
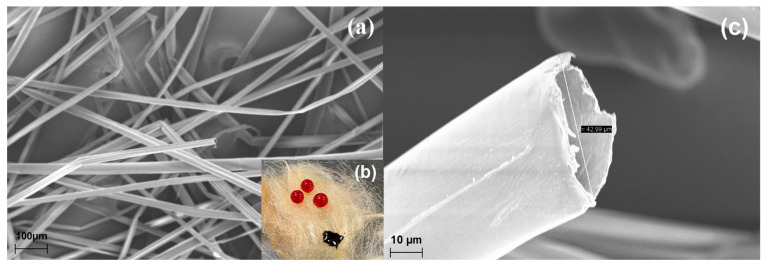
Micrograph obtained by the (**a**) SEM-FEG 100×, (**b**) digital image of the wettability test, and (**c**) SEM-FEG 1000× in CPNaOHT.

**Figure 9 polymers-13-03285-f009:**
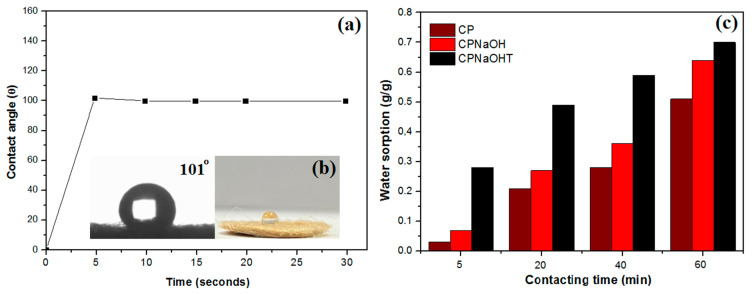
(**a**) CPNaOHT water contact angle test. (**b**) Digital image of tablet CPNaOHT used to measure the contact angle and (**c**) CP, CPNaOH, and CPNaOHT water sorption varying fiber/oil contact time from 5 to 60 min.

**Figure 10 polymers-13-03285-f010:**
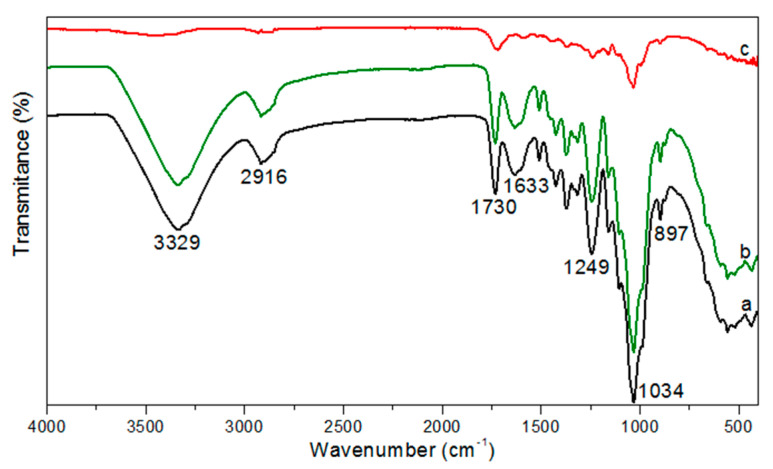
FTIR spectra (**a**) CP; (**b**) CPNaOH; (**c**) CPNaOHT.

**Figure 11 polymers-13-03285-f011:**
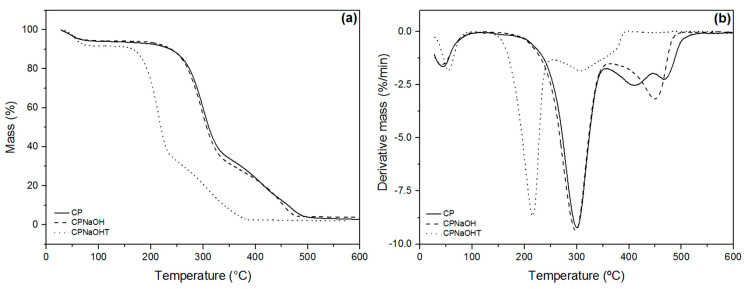
Thermogravimetry (**a**) and Derivative Thermogravimetry (**b**) curves of CP, CPNaOH, and CPNaOHT.

**Table 1 polymers-13-03285-t001:** Factors and levels used in experimental design his is a table.

Factor	Symbol	Level	
		−1	1
Viscosity (cP)	*X* _1_	2.25	73.64
Contact time (h)	*X* _2_	1	24
System	*X* _3_	O	O/W

**Table 2 polymers-13-03285-t002:** Experimental results and calculated results by the models his is a table.

*X* _1_	*X* _2_	*X* _3_	Experimental (g/g)	Calculated (g/g)
−1	−1	−1	95.45	95.44
−1	−1	−1	95.42	
1	−1	−1	92.88	92.66
1	−1	−1	92.44	
−1	1	−1	88.61	88.32
−1	1	−1	88.04	
1	1	−1	121.40	120.91
1	1	−1	120.41	
−1	−1	1	98.87	98.86
−1	−1	1	98.85	
1	−1	1	123.36	123.84
1	−1	1	124.31	
−1	1	1	84.51	84.31
−1	1	1	84.10	
1	1	1	190.14	190.32
1	1	1	190.50	

**Table 3 polymers-13-03285-t003:** Analysis of Variance (ANOVA).

Source	Sum of Squares	DF1	Mean Squares	F	*p*
(1) Viscosity (cP)	6463.76	1	6463.758	38,302.33	5.20 × 10^−16^
(2) Contact time (h)	1334.62	1	1334.624	7908.59	2.85 × 10^−13^
(3) System	2499.75	1	2499.750	14,812.78	2.32 × 10^−14^
1 by 2	3386.95	1	3386.949	20,070.07	6.89 × 10^−15^
1 by 3	2559.60	1	2559.601	15,167.44	2.11 × 10^−14^
2 by 3	237.08	1	237.083	1404.88	2.82 × 10^−10^
1*2*3	521.78	1	521.780	3091.91	1.21 × 10^−11^
Regression	17,003.54	7	2429.078	Significant and predictive	
Residuals	1.35	8	0.168756		
Total	17,004.89	15			
R-sqr = 0.99992	R-adj = 0.99985	F_calc_ = 14394		F_calc_/F_tab_ = 4112.572	

1-DF—Degree of freedom; R-sqr—Correlation coefficient; R-sqr—R adjusted; F_tab_ = 3.5.

## Data Availability

Not applicable.

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
