# Peer review of "Combined Treatment (Alkali + Thermal) of Calotropis procera Fiber for Removal of Petroleum Hydrocarbons in Cases of Oil Spill"

_polymers, 2021, doi:10.3390/polym13193285_

Round 1
Reviewer 1 Report
In order to publish this paper, it needs to solve many problems, as follows:
1- The whole text must be in English and not in combination with the mother tongue of the authors (chapters 2.2; 2.3 and 3.3);
2- The treatment with NaOH (at room temperature) that the authors perform is similar to the mercerization treatment on cotton. The phenomenon of swelling occurs when the lumen grows so much that it almost overlaps with the secondary wall.
When using a temperature higher than 100⁰C, this significantly degrade the PC fibers; this degradation is confirmed by the yellow-brown color after treatment and by Figs. 9 and 10.
The use of high values for the two treatment parameters (NaOH concentration and temperature) inevitably leads to the degradation of any cellulosic fiber, including PC fiber.
3- Curious is that after this treatment with NaOH which usually leads to better water absorption, the authors get the opposite; you should find the explanation of this phenomenon and develop it further in the article. Fig. 10b indicates a more pronounced dehydration step (between 0 and 100⁰C) of the CPNaOHT fiber.
4- Specify if the CPNaOHT fibers have been washed after the treatment! It would be advisable to compare the contact angles on treated and then washed samples respectively on treated samples without being washed later.
5- Compared to cotton, what is the content of morphological companions (lignin, hemicellulose, wax) that confers hydrophobia?
Reviewer 2 Report
This is a useful manuscript. In my opinion, the main result is nicely presented in Figure 7. I recommend to revise the manuscript, reduce its size twice. Minor technical details of the study can be placed in the Supplement Section. The Introduction section should be reduced as well: it is not a review of the literature but a formulation of a task.
The presentation is soppy. The authors did not bother reading the final manuscript.
Lines 101-113, 284 are written in Portuguese!
Some specific minor comments:
line 3: Should be Oil (capital O)
line 18: CPNaOAT is not defined
line 17: All IR nowadays are FT. FT should be omitted here and below
line 304: Here and in other places in the paper is a dimention g/g. It should be replaced by % or by mass/mass in the same units (kg, ton, lb, etc.)
lines 13 and 379: Calotropis provera should be written in the same font in the manuscript. Probably italic should be the best.
Reviewer 3 Report
This work is good reminder for all chemist who worked in membrane research. Various research conducted for removal and purification technologies of oil- spill. Still the issues are not solved completely. I appreciate authors for the current manuscript, using Calotropics procera fiber for treating oil spill with combination of alkali and thermal. And also added the interesting experimental data’s. However, the outcome of the research is not much effective, and found many factors is not considered in this work. I have few questions and suggestions given below;
- Authors should try the real sample collection that would enhance the current manuscript.
- I surprised why authors chose the sorption technique?
- Is there any pretreatment necessary for before sorption?
- Figures and tables title descriptions in manuscript is not enough. Provide more information. Align it properly.
- Did author optimize the temperature for treatment?
- Avoid the typo errors and unnecessary space in the written manuscript.
- Authors must need to incorporate future prospective of the presented work in the conclusion part of the manuscript.
Author Response
Por favor, verifique o anexo.
